# Performance Testing of Micro-Electromechanical Acceleration Sensors for Pavement Vibration Monitoring

**DOI:** 10.3390/mi14010153

**Published:** 2023-01-07

**Authors:** Zhoujing Ye, Ya Wei, Biyu Yang, Linbing Wang

**Affiliations:** 1National Center for Materials Service Safety, University of Science and Technology Beijing, Beijing 100083, China; 2Department of Civil Engineering, Tsinghua University, Beijing 100084, China; 3Zhaotong Highway Investment Development Co., Ltd., Zhaotong 657099, China; 4Highway School, Chang’an University, Xi’an 710054, China; 5School of Environmental, Civil, Agricultural and Mechanical Engineering, University of Georgia, Athens, GA 30602, USA

**Keywords:** pavement vibration, smart road, MEMS accelerometer, sensing performance, comparative evaluation

## Abstract

Pavement vibration monitoring under vehicle loads can be used to acquire traffic information and assess the health of pavement structures, which contributes to smart road construction. However, the effectiveness of monitoring is closely related to sensor performance. In order to select the suitable acceleration sensor for pavement vibration monitoring, a printed circuit board (PCB) with three MEMS (micro-electromechanical) accelerometer chips (VS1002, MS9001, and ADXL355) is developed in this paper, and the circuit design and software development of the PCB are completed. The experimental design and comparative testing of the sensing performance of the three MEMS accelerometer chips, in terms of sensitivity, linearity, noise, resolution, frequency response, and temperature drift, were conducted. The results show that the dynamic and static calibration methods of the sensitivity test had similar results. The influence of gravitational acceleration should be considered when selecting the range of the accelerometer to avoid the phenomenon of over-range. The VS1002 has the highest sensitivity and resolution under 3.3 V standard voltage supply, as well as the best overall performance. The ADXL355 is virtually temperature-independent in the temperature range from −20 °C to 60 °C, while the voltage reference values output by the VS1002 and MS9001 vary linearly with temperature. This research contributes to the development of acceleration sensors with high precision and long life for pavement vibration monitoring.

## 1. Introduction

In order to ensure the efficient, safe and intelligent operation of road-traffic systems, more and more monitoring technologies are applied, and pavement dynamic response monitoring is an essential part. By monitoring and analyzing the pavement dynamic response data under the action of vehicle loads, traffic information and the service state of the pavement structure can be obtained, which provides data support for traffic control and road maintenance [1].

When monitoring pavement dynamic response, the quantities monitored are often stress, strain, displacement, or bending. In recent years, accelerometers have been applied to pavement vibration monitoring under vehicle loading with an improvement in accelerometer performance, such as high sensitivity, low power consumption, and small size [2]. Researchers are already using different acceleration sensors in road engineering to monitor traffic and structural health, as summarized in Table 1.

As shown in Table 1, the acceleration sensors used in road engineering mainly include MEMS, IEPE (integrated electronics piezo-electric), and optical fiber. Among them, IEPE enables high-precision monitoring, which is suitable for roadside deployment. Fiber-optic sensors are low-cost and durable, which are currently used for the vibration monitoring of concrete pavement slabs. However, both of them lack integration and need to be equipped with dedicated data acquisition systems, which leads to high monitoring costs and high-power supply energy consumption. Compared to other sensors, MEMS accelerometers have many advantages, including small size, low power consumption, high accuracy, high reliability, and low cost. The small size of MEMS sensors allows for better integration and packaging, enabling integration with electronic components such as CPUs for edge computing, and can be embedded in the pavement without damaging the road structure. The low power consumption of MEMS sensors allows for self-powering using new energy sources such as solar, wind, and piezoelectric energy, and can greatly save power supply costs for road monitoring in remote areas without the need for cables. The high accuracy and high reliability of MEMS sensors make them suitable for long-term monitoring in harsh service environments. The low cost of MEMS sensors allows for low-cost deployment at multiple points in road structures, making wide-scale use of MEMS sensors possible. In recent years, with the improvement in the performance of MEMS acceleration sensors, MEMS-based acceleration sensors have the capability of real-time acquisition, processing, and analysis. The amount of collected data provides conditions for data-driven pavement vibration monitoring.

However, the pavement materials, structures, and the external environment affect the traffic-induced pavement vibration. The pavement vibration signal is complex and fluctuates randomly, and the vibration amplitude drops rapidly. The high-precision acceleration sensors are needed for pavement vibration monitoring. Nevertheless, there are many kinds of MEMS vibration sensors with different performance parameters. It is necessary to compare and test different MEMS accelerometer chips to analyze the influence of different factors on pavement vibration signals, which can guide the development and application of acceleration sensors in a smart road.

## 2. Acceleration Sensor Development

### 2.1. Hardware Composition

In order to compare the performance of three different MEMS accelerometer chips (MS9001, VS1002, and ADXL355), a sensor, PCB, with different MEMS acceleration chips was developed, as shown in Figure 1.

The main components of PCB include ultra-low-power CPU (STM32F103C6T6A), three MEMS accelerometer chips (VS1002, ADXL355, MS9001), an analog-to-digital converter (AD7172), a buck regulator chip (AMS1117-3.3), and a 485 communication chip (MAX3485ED). Among them, MS9001 and VS1002 output digital signals through the AD converter. The CPU calculates and processes the digital signals and communicates with the host computer through the 485 communication interface. The ADXL355 directly outputs digital signals without the AD converter. The PCB is powered by 5 V USB and depressurized to 3.3 V by AMS1117-3.3 to provide CPU power for the MEMS accelerometer chip. The main functions and datasheet links of the utilized components are shown in Table 2.

### 2.2. Circuit Design

The ADXL355 is a triaxial accelerometer with a range of ±2 g. The pin configuration of the ADXL355 is shown in Figure 2. To ensure a stable 3.3 V supply voltage, a single 100 nF decoupling capacitor was designed in the power port to eliminate supply noise.

MS9001 and VS1002 are high-precision single-axis accelerometers with measuring ranges of ±1 g and ±2 g, respectively. They can output analog signals and convert them into digital signals via AD. The pin configuration of MS9001 and VS1002 is shown in Figure 3.

Three 1 µF capacitors were designed at the power supply to eliminate the power supply noise and ensure the power supply stability. In order to effectively compare the sensing performance of different accelerometer chips, the voltage output terminal was not filtered, and the direct output was adopted. The power supply voltages were both 3.3 V.

### 2.3. Data Acquisition

In order to acquire the original data and test the sensor performance, Visual Studio (VS) was used to design the visual acquisition interface, as shown in Figure 4. VS supports various programming languages and provides a wealth of development tools to help developers easily develop Windows applications, web applications, mobile applications, and more.

The visual acquisition interface of the serial port of the host computer was written in the C# programming language, which has the essential functions of receiving, visualizing, and storing data. The digital signal of the acceleration sensor was output through the serial port. The communication between the host computer and the acceleration sensor was realized by a USB to the serial port. The baud rate of serial communication was set to 115,200 bit/s. This is a commonly used baud rate for serial communication, and it is effective in transmitting data from multiple MEMS accelerometer chips while maintaining a high level of data transmission quality and reducing the risk of interference. The high baud rate allows the host computer to effectively process the data without causing any data loss or degradation in data transmission quality.

## 3. Experimental Design

The vibration frequency of traffic-induced pavement is generally 1–50 Hz, and the peak vibration is within 1 g [28,29]. Therefore, the performance parameters of the accelerometers were tested within the range of ±1 g. The sampling frequency was set as 500 Hz. In order to effectively test and evaluate the accelerometer performance parameters, including sensitivity, linearity, noise, resolution, frequency response, and temperature drift, a corresponding test scheme was designed in this study. In actuality, the datasheet for an accelerometer chip provides relevant performance parameters. However, different manufacturers provide different conditions for these parameters, making it difficult to compare them effectively. For example, the supply voltage for the ADXL355 and the VS1002 is 3.3 V, while the supply voltage for the MS9002 is 5 V. Different supply voltages will affect the values of the performance parameters. In addition, the performance parameters given in the datasheet are typical values, and the sensor’s performance parameters may change depending on the application and circuit design. Therefore, by studying and testing these characteristics in more detail, this study can provide valuable insights and recommendations to engineers and designers who are using or considering using these accelerometers in their products.

### 3.1. Sensitivity Test

Sensitivity is the ratio between the change in output quantity (dv) and the change in the input quantity (da). It is the slope of the input–output characteristic curve, as shown in Equation (1).
(1)$$Sensitivty=\frac{dv}{da}$$

For linear sensors, the sensitivity is a constant, and both dynamic and static calibration tests can be obtained.

(1)Dynamic calibration test

The shaker used for this calibration experiment is the IPA180L/H1248A electrodynamic vibration test system with a rated frequency of 2–2500 Hz, as shown in Figure 5.

Specific practical steps are as follows:1.The measured sensor and the standard sensor were bolted to the shaking table. Both were located in the center area of the shaker to ensure that they were subjected to the same vibration amplitude. The accuracy of the calibration value was guaranteed by simultaneous movement.2.The vibration frequency of the shaker was controlled to stay at 10 Hz. The acceleration amplitude of 0.1 g sine wave was output. The voltage signal output of the acceleration sensor at this time was recorded.3.With the acceleration amplitude of 0.1 g as the increasing quantity, the amplitude gradually increased until it reached 1 g. Each acceleration input corresponding to the sensor output voltage signal was recorded in turn.(2)Static calibration test

The correspondence between the sensitive axis direction of the MEMS accelerometer and the direction of gravitational acceleration was used for calibration, as shown in Figure 6.

When the positive direction of the sensor’s sensitive axis coincided with the direction of the gravitational acceleration, the input acceleration was considered to be 1 g. When the positive direction of the sensor’s sensitive axis was opposite to the direction of gravitational acceleration, the input acceleration was considered to be −1 g. When the direction of the sensor’s sensitive axis was perpendicular to the direction of gravitational acceleration, the input acceleration was considered to be 0 g. Therefore, the sensor output voltage signal was obtained for 1 g, −1 g, and 0 g acceleration.

### 3.2. Noise and Resolution Test

The output signal of the acceleration sensor contains circuit noise and environmental noise. In order to truly reflect the noise level of the internal system of the sensor, the acceleration sensor was placed on the anti-vibration table, and the direction of its sensitive axis was perpendicular to the direction of gravity acceleration. At this time, the acceleration input in the direction of the sensitive axis was 0 g, and the voltage data output by the acceleration sensor were collected.

Standard deviation is a statistical index to measure the deviation between each data value in the data set and the mean value, reflecting the degree of dispersion of data distribution, as shown in Equation (2). The smaller the standard deviation is, the less noise the sensor has. Therefore, the standard deviation σ can be used to represent the noise level of the acceleration sensor [30].
(2)$$\sigma =\sqrt{\frac{\sum (x_{i}-\overset{¯}{x}{)}^{2}}{N-1}}$$

Resolution is the minor change that the sensor can detect and will only sense if the input is higher than the resolution. It is generally considered that the input quantity can be distinguished when it is greater than the system noise. Equation (3) can be used to calculate the resolution of the acceleration sensor.
(3)$$Resolution=\frac{V_{noise}}{Sensitivity}$$

### 3.3. Frequency Response Test

When the input quantity is not fixed but periodically changes, the output of the sensor will also occur with the periodic change of the input quantity, and the frequency of both remains unchanged. However, the output amplitude will vary with the frequency. This variation in the output amplitude with frequency is called the amplitude–frequency characteristic of the signal.

The frequency response range is an important performance index of the sensor. When the signal frequency exceeds this range, the sensor output signal will be distorted. A larger frequency response range means that the sensor can be applied to a broader range of fields. Therefore, it was necessary to test the frequency response range of the accelerometer. The frequency response range could be obtained by testing the amplitude–frequency characteristic curve of the accelerometer. Since the pavement vibration under vehicle load was mainly concentrated in the low frequency range, the vibration amplitude was small. Therefore, the practical steps were set as follows:The measured sensor and the standard sensor were bolted on the shaker.The shaker was controlled to output a sine wave with a constant acceleration amplitude of 0.2 g and frequencies of 2 Hz, 3 Hz, 4 Hz, 6 Hz, 8 Hz, 12 Hz, 16 Hz, 20 Hz, 28 Hz, 32 Hz, 40 Hz, 48 Hz, 56 Hz, 64 Hz, 80 Hz, 96 Hz, 112 Hz, and 128 Hz signals. The output voltage signal of each frequency acceleration sensor was recorded successively.Real-time monitoring data were collected for frequency response range analysis.

### 3.4. Temperature Drift Test

The change in sensor output due to temperature change is called the temperature drift phenomenon. It is important to carefully consider the effects of temperature on the instrumentation system and to implement appropriate temperature compensation measures in order to ensure the accuracy and reliability of the measurements. In order to test the variation in acceleration sensor output with temperature, a temperature control experiment box (DM1000-C15-ESS) was used for testing, and the experimental setup is shown in Figure 7.

Specific practical steps were as follows.

The acceleration sensor was placed horizontally in the temperature control box, fixed with the straps, and connected with the external computer. The sensor was positioned in the center of the box to facilitate heat exchange and ensure accurate temperature control adequately. The temperature in the box was obtained through the temperature sensor on the acceleration sensor (TMP101NA/3K).After adjusting the temperature inside the temperature control box to −20 °C and keeping the constant temperature, the temperature was set to 0 °C, 20 °C, 40 °C, and 60 °C in sequence. The acceleration sensor’s output voltage and temperature data were collected in real-time.The temperature control box was closed to allow it to cool naturally to room temperature. The output signal of the acceleration sensor was collected continuously.

## 4. Experimental Results

### 4.1. Performance Parameters Analysis

#### 4.1.1. Sensitivity

The shaker output acceleration values of 0.1 g, 0.5 g, and 1 g of the three accelerometer chips in the time domain and frequency domain, as shown in Figure 8.

In Figure 8a, the waveform characteristics of the ADXL355 output signal are not prominent under small vibration amplitude, which is affected by noise. In contrast, the output signal of MS9001 has obvious waveform characteristics and small noise, which is suitable for the monitoring of small amplitude. Since the voltage signal output from the sensor contains environmental noise, the vibration signal is transformed from the time domain to the frequency domain using the fast Fourier transform method. From the frequency domain signal, it can be seen that there was high-frequency noise of 150–200 Hz in the sensor output signal, among which the noise signal of ADXL355 was the most obvious and the noise signal of MS9001 was the least. Moreover, the acceleration signal had a period of 0.1 s, which was consistent with the shaker vibration frequency of 10 Hz, and the corresponding amplitude was the maximum at 10 Hz.

In Figure 8b,c, with the increase in vibration amplitude, the waveform characteristics of the signals output from the three accelerometer chips are apparent in the time domain. However, the waveform output from the MS9001 shows a clipping phenomenon, because the measuring range of MS9001 is ±1 g, and the sensitive axis of MS9001 is prone to the over-range phenomenon in the direction of gravitational acceleration due to the effect of gravitational acceleration.

Therefore, the sensitivity of VS1002 and ADXL355 was analyzed using the dynamic calibration test, while the sensitivity of MS9001 was calibrated using a static calibration test. The maximum amplitude of VS1002 and ADXL355 at 10 Hz was used as the output value under the corresponding acceleration input. For example, the output values of VS1002 at 0.1 g, 0.5 g, and 1.0 g were 130.6 mV, 677.2 mV, and 1367.0 mV. MS9001 was statically calibrated by taking the median output voltage of its sensitivity axis upward, corresponding to −1 g input; sensitivity axis horizontal, corresponding to 0 g input; and sensitivity axis downward, corresponding to 1 g input. The resulting relationship between the output voltage and the input acceleration of different accelerometer chips is shown in Figure 9.

As shown in Figure 9, the linear relationship between the output voltage and input acceleration of the three accelerometer chips (R2 = 0.9) is satisfied. The sensitivity of VS1002, MS9001, and ADXL355 is the slope of the fitted straight line. The sensitivity of VS1002 and ADXL355-Z (the *Z*-axis indicates vertical acceleration) at 10 Hz frequency was about 1371.1 mV/g and 393.1 mV/g. The difference with the static calibration results (about 1358.9 mV/g and 395.9 mV/g) was less than 1%. The sensitivity of the MS9001 was approximately 1278 mV/g. It can be seen that VS1002 had the highest sensitivity at 3.3 V supply voltage, and the calibration results of dynamic and static sensitivity were similar.

#### 4.1.2. Noise and Resolution

The output voltage data were collected when the acceleration input was 0 g (i.e., the direction of the sensor’s sensitive axis was perpendicular to the acceleration of gravity), as shown in Figure 10.

In Figure 10, the VS1002 output signal has the smallest fluctuation range within 1 mV, followed by the ADXL355 and MS9001. Because the sensitive axis cannot avoid the influence of the earth’s gravitational acceleration, the reference value is not 0 mV. The noise was calculated using Equation (2), and the resolution was calculated using Equation (3), as shown in Table 3.

#### 4.1.3. Frequency Response Curve

The frequency response of the three types of accelerometer chips is shown in Figure 11.

In Figure 11a, the sensitivity of the three accelerometer chips maintained a smooth trend at vibration frequencies of 2–128 Hz. In Figure 11b, the upper and lower quartiles of the VS1002 sensitivity were 1363 mV/g to 1373 mV/g. In addition, the VS1001 sensitivity at a low frequency of 2 Hz was 1402 mV/g, which was an outlier in its data sample. Compared to the VS1002, the upper quartile (1320 mV/g) and lower quartile (1303 mV/g) of the MS9001 sensitivity differed by a smaller amount, but there were four outlier values in the data sample, indicating that the stability of its sensitivity at low frequencies was not as good as that of the VS1002. The difference between the upper and lower quartiles of the ADXL355-Z sensitivity was only 4 mV/g, and its sensitivity output at low frequencies was the most stable, which is related to its built-in filtering circuit and lower sensitivity. Therefore, under the premise of meeting the sensitivity requirements, the ADXL355-Z has the best stability at low frequencies of 2–128 Hz. This indicates that the VS1002 and MS9001 can be further improved by designing a filtering circuit.

### 4.2. Temperature Drift Evaluation

In the temperature drift test experiment, the temperature control box will produce stable motor vibration noise during operation, so the signal of the acceleration sensor will fluctuate, as shown in Figure 12.

In Figure 12, in the temperature-control-box operating environment, the VS1002 and MS9001 had a fluctuation of about ±250 mV, while the fluctuation range of ADXL355 was smaller, about ±75 mV. This is due to the lower sensitivity and the built-in filtering circuit of the ADXL355, which can filter out environmental noise. According to the datasheet of ADXL355, the ADXL355 can directly output digital signals without needing an external analog-to-digital converter. A three-axis sensor, temp sensor, ADC, analog filter, and digital filter are integrated and packaged in ADXL355. The analog, low-pass, anti-aliasing filter in the ADXL355 provides a fixed bandwidth of approximately 1.5 kHz, which is where the output response is attenuated by approximately 50%. The shape of the filter response in the frequency domain is that of a sinc3 filter. The ADXL355 provides further digital-filtering options to maintain excellent noise performance at various output data rates.

The probability density analysis of the output signal showed that the output voltage values of the three sensor chips were distributed as Gaussian distribution, and the R^2^ of the Gaussian fitting curve was more significant than 0.9. The output voltage can be considered as the sum of a constant value ($v_{c}$) and noise value ($v_{n}$), as shown in Equation (4).
(4)$$v=v_{c}+v_{n}$$

The noise value can be approximately fitted using Gaussian distribution. Based on the minimum root mean square (RMS), the constant value can be calculated using the regression analysis technique. For the acceleration output signal within 10 s, the $v_{c}$ value was taken as the actual output of the acceleration sensor within the unit time (10 s). The output value of the acceleration sensor was continuously recorded during the temperature change. Figure 13 shows the temperature drift of different accelerometer chips. Figure 13a presents the process of output $v_{c}$ of three accelerometer chips with temperature, i.e., the temperature drift process. Figure 13b presents the effect of temperature on the reference value of the accelerometer chip output during the temperature stabilization phase.

As shown in Figure 13, the output reference value of ADXL355 was not affected by temperature, while the output reference value of VS1002 and MS9002 varied linearly with temperature. Moreover, the reference value is positively correlated with temperature. The MS9002 was more affected by temperature than the VS1002. The reference value of the output voltage of the MS9002 varied with temperature at a rate of 0.255 mV/°C, while that of the VS1002 varied with temperature at a rate of 0.102 mV/°C. The output reference value of the sensor changed significantly during the warming or cooling phase, and the trend was similar to the temperature change curve.

## 5. Conclusions

In this paper, the performance testing and comparative evaluation of MEMS high-precision acceleration sensors are carried out, contributing to the development and application of acceleration sensors in intelligent road engineering. The main conclusions are as follows:(1)PCBs with three accelerometer chips, VS1002, MS9001, and ADXL355, are developed, and circuit design and software development are completed to realize the real-time output and visualization of vibration data. The sensor sensitivity, linearity, noise, resolution, frequency response range, and temperature drift test are designed.(2)The range of MS9001 is ±1 g. Due to the acceleration of gravity, the over-range phenomenon occurs after loading 0.3 g acceleration. Thus, it cannot be applied to the pavement vibration monitoring on the wheel path. However, it can be placed on the roadside to monitor the pavement’s small amplitude fluctuations.(3)The VS1002 has the best performance indexes. Under the supply voltage of 3.3 V, the sensitivity of VS1002 can reach about 1371.1 mV/g, the noise can reach 0.231 mV, and the resolution can reach 0.169 mg, which can be used as the first choice for the high-precision monitoring of pavement vibration.(4)The ADXL355 has low noise and sensitivity; the resolution is 0.655 mg. In the temperature range from −20 °C to 60 °C, the ADXL355 signal output is virtually temperature-independent. The chip is low-cost and suitable for pavement vibration monitoring at higher-density deployments.

The performance test of MEMS accelerometers will help the open-source development of modules/add-ons for smart roads and promote the application of MEMS sensor technology in the transportation industry. Furthermore, filter circuit design and package optimization will be conducted based on the preferred MEMS accelerometer chip. The variation law of vibration time–frequency domain characteristic parameters in extreme service environments should be expounded. Moreover, sensors with high accuracy and long life will be developed for pavement vibration monitoring.

## Figures and Tables

**Figure 1 micromachines-14-00153-f001:**
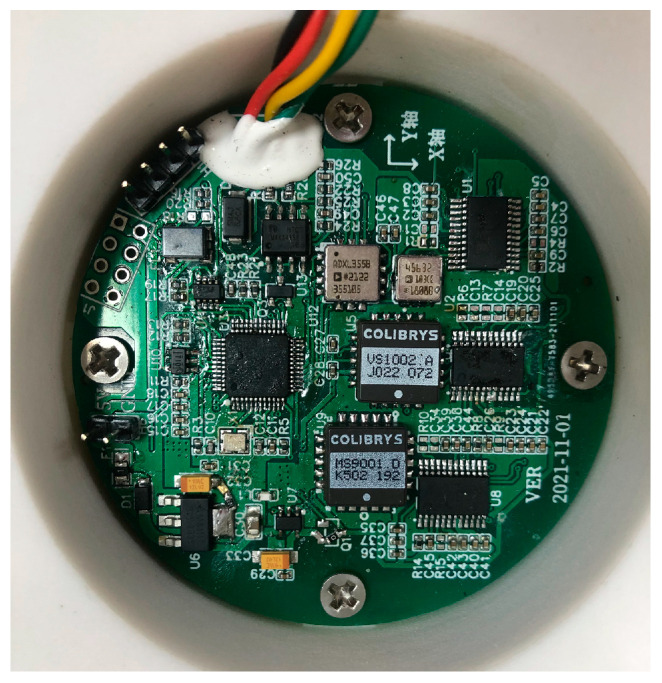
The PCB of the acceleration sensor.

**Figure 2 micromachines-14-00153-f002:**
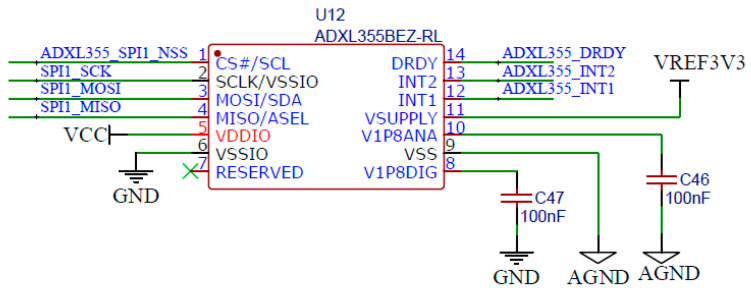
The pin configuration of the ADXL355.

**Figure 3 micromachines-14-00153-f003:**
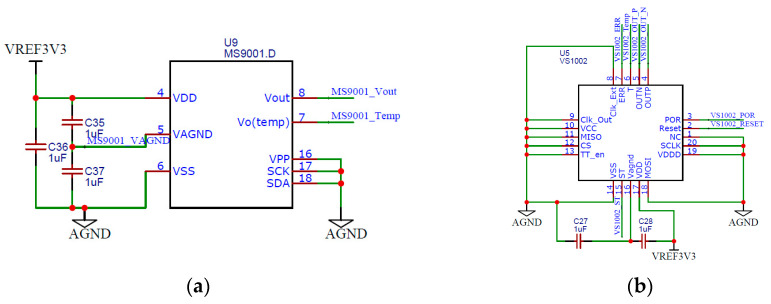
The pin configuration of the MS9001 and VS1002: (**a**) MS9001; (**b**) VS1002.

**Figure 4 micromachines-14-00153-f004:**
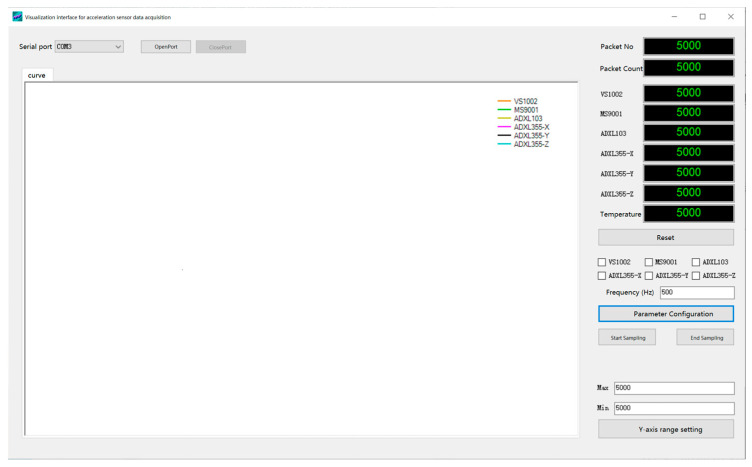
Visualization interface for acceleration sensor data acquisition.

**Figure 5 micromachines-14-00153-f005:**
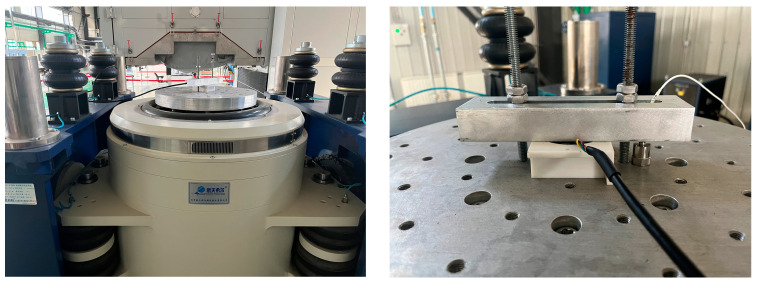
Acceleration sensor dynamic calibration test.

**Figure 6 micromachines-14-00153-f006:**
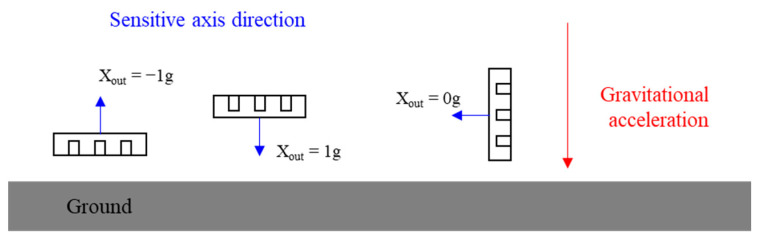
Correspondence between sensitive axis direction and gravitational acceleration direction.

**Figure 7 micromachines-14-00153-f007:**
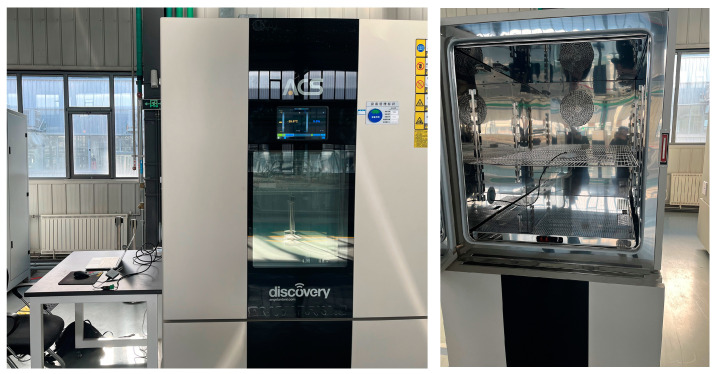
Temperature drift experiment for the acceleration sensor.

**Figure 8 micromachines-14-00153-f008:**
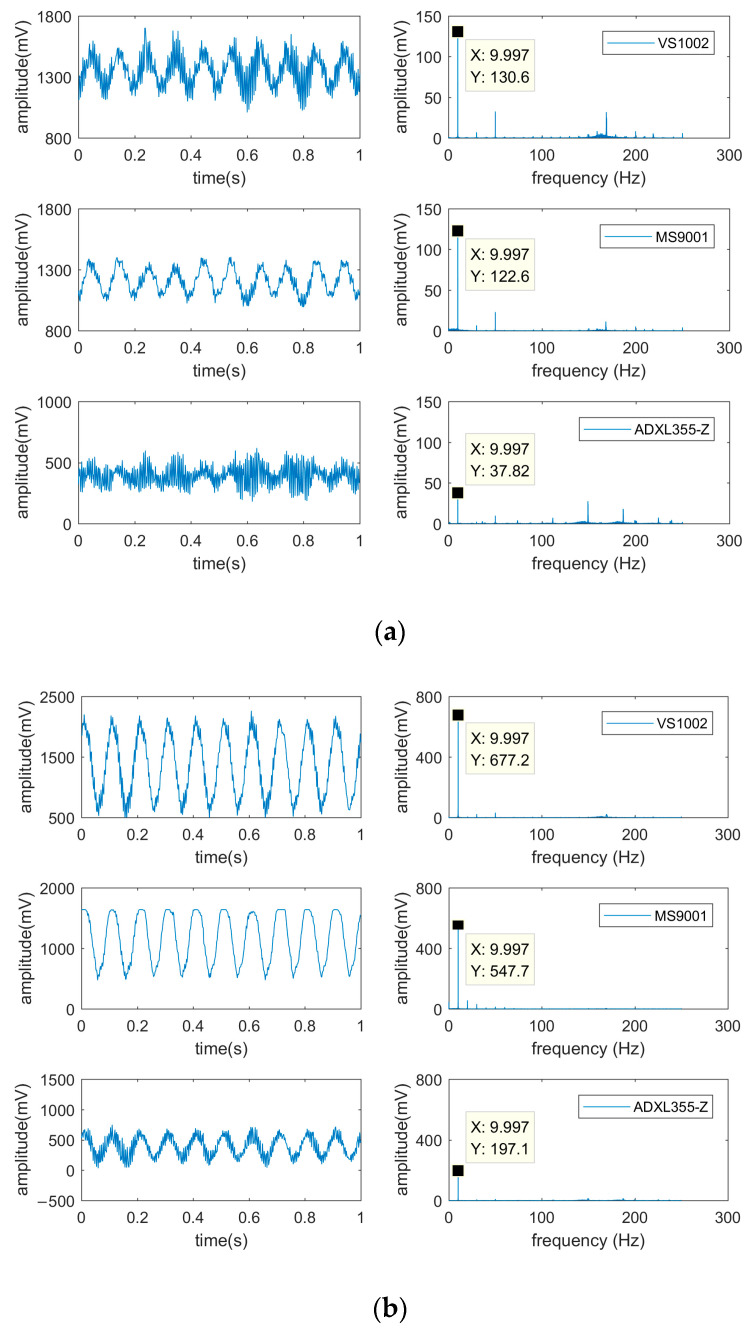
Voltage signal output at different acceleration amplitudes: (**a**) The output acceleration amplitude of the shaker is 0.1 g; (**b**) The output acceleration amplitude of the shaker is 0.5 g; (**c**) The output acceleration amplitude of the shaker is 1.0 g.

**Figure 9 micromachines-14-00153-f009:**
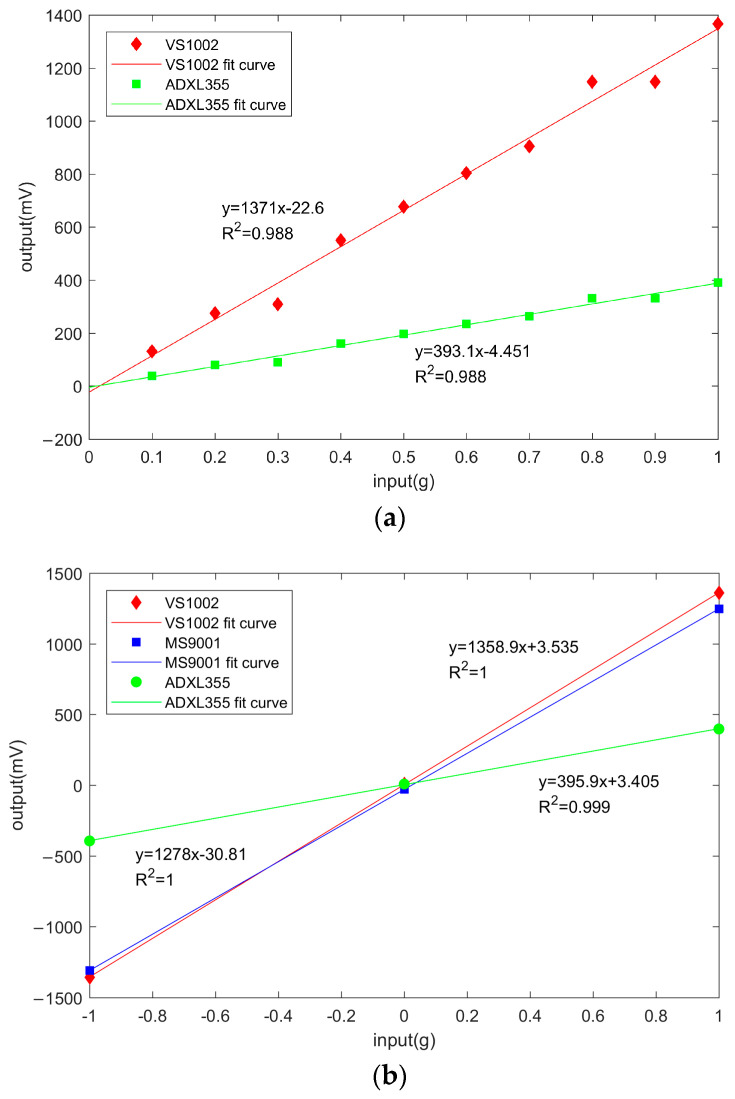
The relationship between the output voltage and input acceleration of different accelerometer chips: (**a**) Dynamic calibration test results; (**b**) Static calibration test results.

**Figure 10 micromachines-14-00153-f010:**
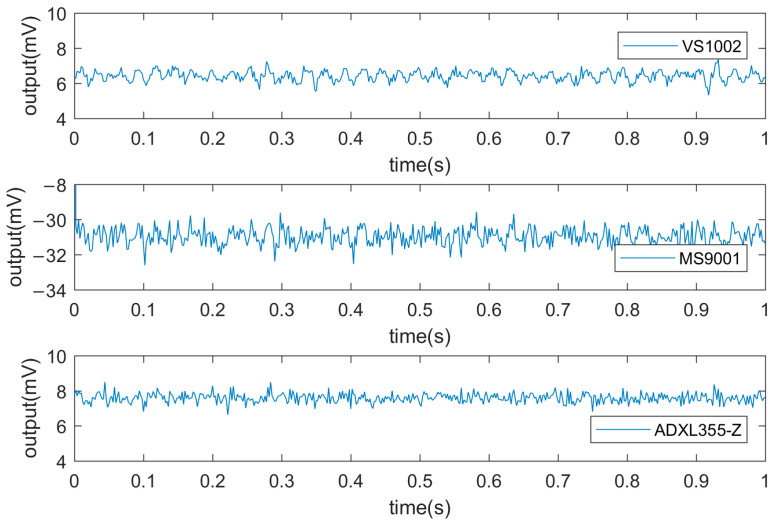
Sensor voltage signal output when acceleration input is 0 g.

**Figure 11 micromachines-14-00153-f011:**
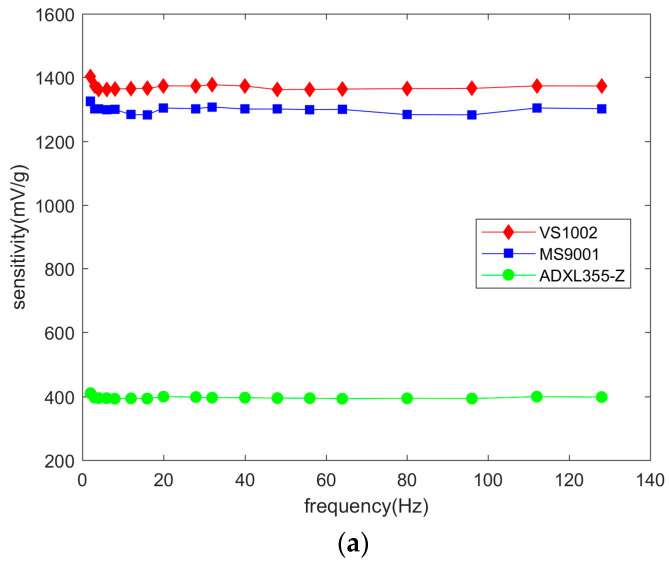
Frequency response of different accelerometer chips: (**a**) Frequency response curves; (**b**) Frequency response box plot.

**Figure 12 micromachines-14-00153-f012:**
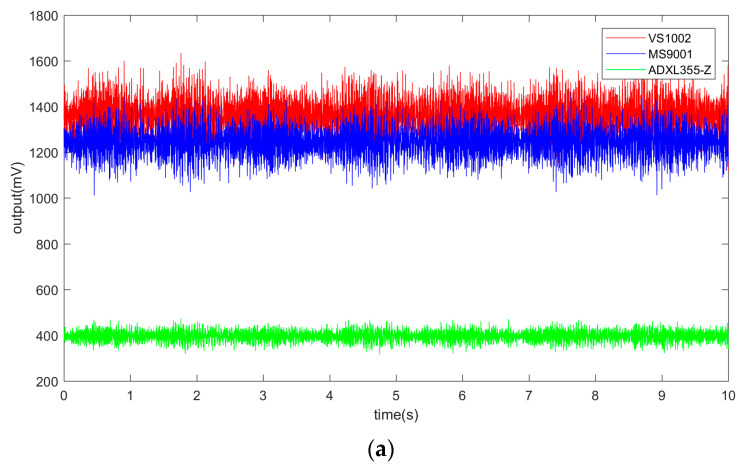
Acceleration sensor output signal during temperature control box operation: (**a**) 10 s of acceleration sensor output signal; (**b**) Probability density distribution of acceleration sensor output values.

**Figure 13 micromachines-14-00153-f013:**
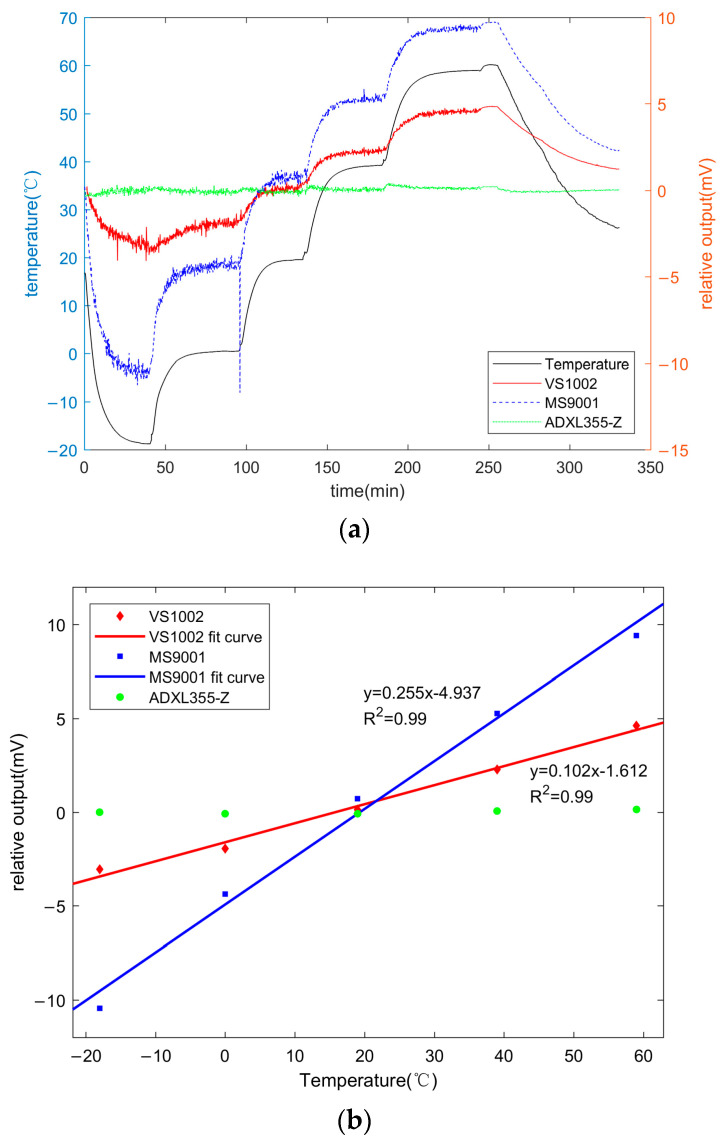
Temperature drift of different accelerometer chips: (**a**) Variation in the reference value of the acceleration sensor with temperature change; (**b**) Effect of temperature on the output reference values of different acceleration sensors.

**Table 1 micromachines-14-00153-t001:** Acceleration sensors used in road engineering.

Name	Supplier	Type	Deployment	Purpose
CEF C3M01 accelerometer	Webrosensor Oy4 as WBS CM301	MEMS	The accelerometer vibration sensors were installed at a relative distance of approximately 45 m on the right side along one of the roads at the training site.	Vehicle counting and classification [3,4]
Road marking units	Self-development	MEMS	A pair of sensor nodes were deployed on both sides of the two-lane, two-way highway. The two sensors on the same side were separated by 6m and worked in pairs for the detection and estimation. The accelerometer can optionally be used to improve the monitoring accuracy.	Vehicle counting and classification [5,6]
G-link-LXRS wireless accelerometer node	LORD Sensing	MEMS	The G-link wireless accelerometer node was installed at the edge of the pavement, with a 1m distance to the wheel path.	Vehicle counting and classification [7]
The wireless accelerometers	Sensys Networks	MEMS	Detection sensors report vehicle detection time, and multiple arrays of vibration sensors report pavement acceleration for load estimation. MS9002 from Colibrys was used due to its low operating voltage and low current consumption.	Vehicle counting, vehicle classification, and vehicle weight estimation [8,9,10]
An acceleration-sensing node	Self-development	MEMS	The acceleration-sensing nodes were installed on the wheel path. Two rows of acceleration-sensing nodes were installed 3.1 m apart.	Vehicle speed, Vehicle type, and abnormal vehicle weight [11,12]
SmartRock	STRDAL	MEMS	SmartRock sensors were installed in an asphalt pavement test section using an accelerated pavement testing (APT) loading facility as well as an in situ pavement to collect the pavement responses.	Vehicle speed and pavement compactness [13,14,15,16]
ICP (integrated circuit piezoelectric) accelerometer	unknown	IEPE	The accelerometer can be mounted at any convenient location. In this paper, it is mounted on the frame under the passenger seat by a magnetic base.	Road surface roughness [17]
Triaxial accelerometer	ADXL335	MEMS	An accelerometer was mounted on each front and rear axle’s hubs with the help of adhesive to be held securely.	Tire pressure [18]
An accelerometer	unknown	IEPE	An accelerometer was installed 40 mm beneath the pavement’s surface in five sections of the circular test track, CTT, in the center line of the wheel track.	Pavement deflection [19]
A high-end single-axis inertial sensor (KB12VD)	Metra Meß (Germany)	IEPE	A field experiment was performed using the KB12VD; it consisted of embedding the sensor in an asphalt pavement and driving a vehicle of known weight and dimensions close to the embedment location.	Vehicle speed, axle weight distribution, and surface layer modulus [20,21]
A wireless accelerometer	unknown	MEMS	The wireless accelerometer was mounted on the HMA slabs at 10 cm off the wheel path for acceleration measurements.	Structural condition assessment of asphalt concrete slab [22]
Triaxial accelerometer	unknown	MEMS	A triaxial accelerometer is adopted to test the pavement vibration acceleration signals caused by the bus.	Structural cracking assessment [23]
Structural wireless test systems (A1005 ± 2 g)	Bridge Diagnostics, Inc	MEMS	The sensors are attached to the runway surface using special glue. There are three measurement lines with a spacing of 15 m. Four acceleration sensors are placed on each measurement line, and the spacing of sensors from the aircraft runway centerline to shoulder direction are 2 m, 2 m, and 3 m, respectively.	Pavement deflection [24]
A distributed optical vibration sensing system (DOVS)	Self-development	Optical Fiber	Pre-fixing the fiber to the reinforcement, then pouring concrete, and high-density placement at the corners and edges of the slab.	Support conditions assessment of concrete pavement slab [25,26,27]

**Table 2 micromachines-14-00153-t002:** The main functions and datasheet links of the utilized components.

Component	Supplier	Function	Datasheet Link
ADXL355	ANALOG DEVICES	Three-axis accelerometer with an operating temperature range of −40 °C to 125 °C and full-scale acceleration of ±2 g	https://pdf1.alldatasheet.com/datasheet-pdf/view/903165/AD/ADXL355.html: accessed on 7 January 2023
MS9001	SAFRAN Colibrys	Single-axis accelerometer with an operating temperature range of −55 °C to 125 °C and full-scale acceleration of ±1 g	https://www.colibrys.com/wp-content/uploads/2015/03/30s-ms9001d-a-12-15.pdf: accessed on 7 January 2023
VS1002	SAFRAN Colibrys	Single-axis accelerometer with an operating temperature range of −55 °C to 125 °C and full-scale acceleration of ±2 g	https://www.colibrys.com/wp-content/uploads/2018/03/30s-vs1000-e-02-18.pdf: accessed on 7 January 2023
STM32F103C6T6A	STMicroelectronics	32-bit high-performance, low-power, general-purpose microcontroller for original equipment control and data processing	https://www.allaboutcircuits.com/electronic-components/datasheet/STM32F103C6T6A--STMicroelectronics/: accessed on 7 January 2023
AD7172	ANALOG DEVICES	24-bit ADC that converts analog signals into digital signals with an operating temperature range of −40 °C to 105 °C	https://www.analog.com/media/en/technical-documentation/data-sheets/ad7172-2.pdf: accessed on 7 January 2023
AMS1117-3.3	ams-OSRAM AG	A voltage regulator that reduces 5 V voltage to 3.3 V to power the processor and sensors	https://pdf1.alldatasheet.com/datasheet-pdf/download/205691/ADMOS/AMS1117-3.3.html: accessed on 7 January 2023

**Table 3 micromachines-14-00153-t003:** Sensitivity, noise, and resolution of different accelerometer chips.

Accelerometer	Sensitivity Tested by Dynamic Calibration (mV/g)	Sensitivity Tested by Static Calibration (mV/g)	Noise (mV)	Resolution (mg)
VS1002	1371.1	1358.9	0.231	0.169
MS9002	--	1278	0.461	0.361
ADXL355-Z	393.1	395.9	0.257	0.655

## Data Availability

The data presented in this study are available upon request from the corresponding author.

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
