# Peer review of "Performance Testing of Micro-Electromechanical Acceleration Sensors for Pavement Vibration Monitoring"

_micromachines, 2023, doi:10.3390/mi14010153_

Round 1

Reviewer 1 Report

The paper discusses about the comparison between three MEMS accelerometer chips to find out the suitable one for pavement vibration monitoring.

There are the following concerns that which authors need to clarify and update:

1.      In the introduction, the authors should clarify more about the motivations and advantages of using MEMS sensors in pavement vibration monitoring.

2.      Authors should emphasize the importance of this research over these accelerometer datasheets (Why is the datasheet not deep enough? Such as noise, resolution, sensitivity, etc.)

3.      Authors should add the references or datasheet links of the utilized components such as MEMS accelerometer chips (VS1002, ADXL355, MS9001), analog-to-digital converter (AD7172), buck regulator chip (AMS1117-3.3), etc.

4.      What is the baud rate of serial communication? Why did the authors choose this rate?

5.      What is the software author used to design the visual acquisition interface?

6.      The equation quality must be improved in Figure 9, which is not clear

7.      From line 237 to 239 indicate the sensitivity data. Authors should use the term’ approximately or about’ since the results have tolerance.

8.      In section 4.1.3, the analysis must be more detailed, authors state that ‘the sensitivity of the three accelerometer chips maintains a smooth trend so can be applied to the monitoring of low-frequency vibration of pavement structures’. The term ‘smooth trend’ is not enough to claim the reason for using these accelerometers. More explanations are needed.   

9.      In line 276 of section 4.2, the authors state that ‘the fluctuation range of ADXL355 is

smaller due to the lower sensitivity and the built-in filtering circuit

of the ADXL355, which can filter out environmental noise’. What is the type of filter here?

Why does only ADXL355 have this filter circuit, and the two other accelerometers do not have any filter? Does this filter create a significant difference in the ADXL355 performance under the noise and frequency response? Does this filter make the output reference value of ADXL355 is not affected by temperature?

10 . In line 298, ‘In Figure 13, The output reference’, ‘The’ must not be in the caption. Authors must also double-check the English grammar and improve the paper format quality. 

Reviewer 2 Report

The authors have presented the topic well. There are a few questions/suggestions as follows.

1. Justify sensor positioning.

2. What was the sampling frequency?

3. What about temperature compensation?

4. How to ensure the robustness of the proposed system (instrumentation) in a heavy noise environment?

5. While discussing results, please write inferences clearly.

6. You may refer to suggested papers to strengthen the literature review and Table 1. ‘Application of Machine Learning for Tool Condition Monitoring in Turning,’ and ‘Tyre Pressure Supervision of Two Wheeler Using Machine Learning' 7. In continuation, you may try connecting your work to the open-design movement reviewed in the paper ‘Overview of contemporary systems driven by open-design movement’. This would open new opportunities for authors and readers as well. You may present this in future scope.

All the best.

Round 2

Reviewer 2 Report

The authors have addressed all my comments positively. Happy new year. All the best.